# Impact of Intraoperative Blood Transfusion on Cerebral Injury in Pediatric Patients Undergoing Congenital Septal Heart Defect Surgery

**DOI:** 10.3390/jcm13206050

**Published:** 2024-10-11

**Authors:** Artem Ivkin, Evgeny Grigoriev, Alena Mikhailova

**Affiliations:** Research Institute for Complex Issues of Cardiovascular Diseases, 650002 Kemerovo, Russia; grigorievev@hotmail.com (E.G.); carfagenez@mail.ru (A.M.)

**Keywords:** children, neuroprotection, systemic inflammatory response, cardiac surgery, transfusion

## Abstract

**Background:** The components of donor blood themselves have the potential to initiate a systemic inflammatory response and exacerbate neuroinflammation, resulting in subsequent cerebral injury. The aim of this study was to establish the role of transfusion in the development of cerebral injury during the correction of congenital heart defects in children. **Material and Methods:** A total of 78 patients aged from 1 to 78 months, with body weights ranging from 3.3 to 21.5 kg, were investigated. Biomarkers of cerebral injury and systemic inflammatory response were studied at three time points. First: prior to the surgical intervention. Second: after the completion of cardiopulmonary bypass. Third: 16 h after the conclusion of the surgery. **Results:** The strongest correlation was found for S-100-β protein with the volume of transfusion at the second (Rho = 0.48, *p* = 0.00065) and third time points (Rho = 0.36, *p* = 0.01330). Neuron-specific enolase demonstrated a similar trend: Rho = 0.41 and *p* = 0.00421 after the completion of cardiopulmonary bypass. **Conclusions:** The use of red blood cell suspension and its dosage per kilogram of body weight correlated with the biomarkers of cerebral injury and systemic inflammatory response with moderate to significant strength.

## 1. Introduction

Contemporary cardiac surgery and cardiac anesthesia have reached such a high level of development that any surgical intervention must consider not only the preservation of the patient’s life and health but also the quality of their life. The cognitive sphere is directly related to this trend, and it is essential to make every effort to maintain it at the preoperative level, including in pediatric patients undergoing the correction of congenital heart defects (CHD). However, numerous studies have shown that cardiac surgical interventions in children are associated with a high incidence of postoperative cognitive disturbances. Postoperative delirium (POD) occurs in 40–65% of cases [1,2,3,4]. This condition significantly prolongs a patient’s stay in the intensive care unit due to increased needs for mechanical ventilation, sedative medications, and other interventions [1,5]. This prolongs the duration of the length of hospitalization and increases the potential mortality [1,6]. Moreover, the negative impact of POD is not limited to these effects, as it has been shown to have long-term consequences, reducing a child’s cognitive abilities for many months after surgery [7]. Similar long-term outcomes may result not only from POD but also from any cerebral injury that occurs during the intraoperative period [8]. There is evidence of negative behavioral changes in children who have experienced an episode of postoperative delirium. These changes may manifest as attention problems, emotional instability, and difficulties in social interactions. Research indicates that such children may face long-term consequences that affect their psycho-emotional well-being and quality of life [9]. This leads to difficulties in utilizing already acquired skills as well as challenges in learning new information, which is particularly relevant for young children in their early years of life [10,11]. From an economic perspective, the diagnosis of delirium in children increases medical and healthcare costs [12].

When discussing the reasons for the high frequency of cognitive impairments following the correction of CHD in children, it is important to note that cardiac surgeries encompass a wide range of pathological factors that negatively affect the brain and lead to corresponding cerebral injury. The influence of anesthetics, episodes of hypoxia and unstable hemodynamics, the use of sympathomimetic drugs, and the duration of surgery are all typical, yet not unique, to this type of operation [13,14,15]. A distinguishing feature of cardiac surgeries is the use of cardiopulmonary bypass (CPB), under which most procedures take place. The effects associated with CPB that pose risks for cerebral injury include microembolism, the laminar flow of blood, potential episodes of hypothermia, and circulatory arrest [16,17,18]. Furthermore, during any CPB procedure, the patient’s blood comes into contact with the surface of the CPB circuit, which when combined with the aforementioned factors, leads to the initiation and exacerbation of the SIRS [18].

Evidence supports the mechanism by which systemic inflammatory response syndrome (SIRS) contributes to cerebral injury by maintaining neuroinflammation, disrupting the blood–brain barrier, and facilitating the penetration of systemic cytokines through it [19,20]. Furthermore, systemic cytokines stimulate astrocytes to release local cytokines that enhance neuroinflammation. Additionally, systemic cytokines may activate perivascular macrophages in cerebral vessels and promote the infiltration of peripheral monocytes and macrophages into the neurovascular unit, exacerbating its damage [21,22]. In addition, cytokines, particularly interleukin-1β, can induce dysfunction of glymphatic clearance through interaction with astrocytes, thereby impairing the elimination of toxins, cytokines, and chemokines, perpetuating a vicious cycle of neuroinflammation [23]. Therefore, there is a combination of direct damaging factors and those acting indirectly through systemic inflammation, resulting in the subsequent death of the neurovascular unit (Figure 1).

In this context, the application of any methodologies aimed at limiting SIRS in children undergoing the correction of congenital heart defects (CHD) is of particular interest. One potential approach is to restrict the use of donor blood components during the intraoperative and postoperative periods. This issue is especially critical for pediatric patients, as transfusions in cardiac surgery are often conducted preventively to avoid excessive hemodilution [21] due to the mismatch between the child’s circulating blood volume and the priming volume of the CPB machine. Additionally, studies have shown that donor blood components themselves serve as sources of SIRS [24,25]. However, despite the complexity of this issue, the use of donor blood components in CPB procedures is not systematically regulated and remains at the discretion of the anesthesiologist. Currently, a restrictive transfusion strategy predominates worldwide in pediatric patients due to the numerous risks associated with the use of donor blood components. These risks include the potential for infections, acute hemolytic reactions, and the occurrence of Transfusion-Associated Circulatory Overload (TACO) and Transfusion-Related Acute Lung Injury (TRALI) syndromes. However, in cardiac surgery, transfusion remains widely used due to concerns regarding hemodilution during the period of cardiopulmonary bypass [26,27,28]. Therefore, it is pertinent to investigate the extent of the impact of transfusion on the brain and the degree of SIRS in pediatric patients, as well as to identify the relationship between these factors, which was the objective of our study.

## 2. Materials and Methods

### 2.1. Study Design

The study was conducted at the Research Institute of Cardiovascular Surgery. The inclusion criteria for patients in the study were as follows: planned surgical intervention for the correction of congenital heart defects (atrial septal defect or ventricular septal defect) under cardiopulmonary bypass, possession of informed consent for participation in the study signed by the legal representative of the child, age of the child from 1 month to 7 years, and body weight from 3 to 25 kg. Exclusion criteria included: absence of informed consent from the patient and parents for participation in the study; emergency or urgent surgical interventions; presence of clinically significant anemia; hypothermic cardiopulmonary bypass; episodes of desaturation during the perioperative period; presence of other congenital heart defects besides atrial septal defect or ventricular septal defect, as well as their combinations; a history of central nervous system disorders; presence of an established pacemaker; hemodynamic instability requiring preoperative pharmacological and/or mechanical support; any episodes of cerebrovascular accidents in the medical history or during the perioperative period; presence of severe comorbidities that adversely affect psychological and somatic status; acute infection or exacerbation of chronic infection during the perioperative period; associated autoimmune diseases; presence of malignant tumors; and surgical complications in the postoperative period. A total of 78 patients were examined, aged from 1 to 78 months (median 13 (9–23) months), with body weights ranging from 3.3 to 21.5 kg (median 8.7 (6.9–11.0) kg).

The power analysis of the study was performed using the formula: n = (*t*2 × P × Q)/∆2; (t) is the critical value of the Student’s *t*-test at the corresponding significance level, which in this study was set at 0.05; (\Delta) is the allowable margin of error (%); (P) is the proportion of cases exhibiting the studied characteristic (%); and (Q) is the proportion of cases not exhibiting the studied characteristic (100–P). According to the calculations, the sample size should be 196 patients. However, since the effect of limiting red blood cell transfusion on reducing the severity of the SIRS and cerebral injury was significant, a sufficient number of patients included in the study was deemed adequate to demonstrate that this effect was not coincidental. This sample size was sufficient for publishing pilot data, as statistically significant correlations had already been identified. However, we plan to include a calculated number of patients in future studies. This will enable a more accurate calculation of correlation coefficients and allow for a more precise analysis of logistic regression to identify all factors influencing cerebral injury. The study was a prospective observational investigation and was approved by the institutional review board of the Research Institute of Cardiovascular Surgery (protocol No. 17 dated 20 November 2020).

### 2.2. Anesthetic Management

All patients received anesthetic management according to a standardized protocol adopted by the clinic. Upon the patient’s arrival in the operating room, peripheral venous catheterization was performed under local anesthesia. Anesthesia induction was achieved with the administration of propofol at a dose of 2 to 3 mg/kg and fentanyl at 5 µg/kg. For muscle relaxation, bromide of pipecuronium was used at a dose of 0.1 mg/kg. Subsequent procedures included endotracheal intubation, central venous catheterization, radial artery catheterization, and bladder catheterization. At the onset of the surgical intervention, a bolus of fentanyl (5 µg/kg) was administered.

Maintaining anesthesia involved a continuous infusion of propofol at a rate of 2 to 4 mg/kg/h and fentanyl at 5 µg/kg/h using an infusion pump, along with inhalation of sevoflurane at a concentration of 1.0 to 1.5 MAC. Mechanical ventilation was conducted using the Datex-Ohmeda Avans ventilator (General Electric, Chicago, USA) in a semi-closed circuit with SIMV mode under normal ventilation conditions, with the following parameters: FiO_2_ = 0.25–0.3; Vt = 6–8 mL/kg; Pi = 10–15 cm H_2_O; PEEP = 5–8 cm H_2_O; Ti:Te = 1:2. The level of CO_2_ in the exhaled air was monitored.

Assessment of the adequacy of oxygen delivery and consumption by the tissues was conducted using venous blood saturation (SvO^2^), blood lactate levels, and cerebral oximetry measurements (rSO^2^). Additionally, pulse oximetry data (SpO^2^), as well as hemoglobin and hematocrit levels obtained from blood gas analysis, were evaluated.

Any neurological symptoms that arose during the intraoperative and postoperative periods were documented by us.

All patients underwent transthoracic echocardiography before and after the surgery to monitor heart parameters and the success of the surgical correction of the heart defect.

### 2.3. Characteristics of Cardiopulmonary Bypass

Cardiopulmonary bypass (CPB) was conducted according to a standardized protocol adopted by the clinic, utilizing the Maquet HL 20 machine. Membrane oxygenators were employed, including the Baby Fx-05 (Terumo, Tokyo, Japan), Dideco D101 (Sorin Group, Milan Italy), and QUADROX-i Pediatric (Maquet, Germany). The choice of oxygenator depended on the calculated volumetric flow rate for CPB. The priming volume ranged from 300 to 500 mL, and mannitol, sodium bicarbonate, and heparin were added to the priming volume in calculated dosages. A 10% albumin solution was utilized as a colloid at a dosage of 1 g/kg of body weight, while a polyionic solution served as the crystalloid. A leukoreduced red blood cell suspension was used as the erythrocyte-containing component of donor blood in all cases. For n patients with a body weight of less than 7.5 kg, erythrocyte suspension was added to the priming volume before the onset of extracorporeal circulation (ECC) at a dosage of 10–15 mL/kg. In patients with a body weight exceeding 7.5 kg, only the solutions described above were included in the priming volume, without the use of blood components from donors. During CPB, if the hematocrit level dropped below 25% or if venous blood saturation fell below 70%, red blood cell suspension was infused into the CPB circuit at a dosage of 5 to 10 mL/kg, with repeat administration as necessary. After CPB and prior to the end of the operation, transfusion was required for 14 patients, administered at a dose of 5 to 10 mL/kg. No transfusions were given to any patients during the postoperative period.

For cardioplegia, a cooled Custodiol solution was administered at a dosage of 50 mL/kg, with an exposure time of no less than 8 min. The cardioplegic solution was delivered antegradely to the aortic root.

During CPB, ultrafiltration was performed to eliminate excess fluid in the perfusate and prevent hemodilution, utilizing BC 20 plus or BC 60 (Maquet, Germany) plus ultrafiltration columns, depending on the patient’s weight.

In the group receiving red blood cell suspension in the initial priming volume before CPB, modified zero-balance ultrafiltration was performed. Following the completion of CPB, all patients underwent modified ultrafiltration using methods described in the literature, with blood drawn from the arterial cannula and concentrated blood returned through the inferior vena cava cannula.

### 2.4. Applied Markers

The following Cloud-Clone Corp. (Export Processing Zone, Wuhan, China) markers were used to diagnose SIRS: IL-1 (HEA563Hu); IL-6 (HEA079Hu); IL-10 (SEA056Hu); and TNF-α (HEA133Hu) [29,30]. The following markers were used to diagnose NBE damage: S-100-β (SEA567Hu); NSE (SEA537Hu); and GFAP (SEA068Hu) [31,32,33,34]. These markers were selected as the most validated and well-studied, including in the pediatric population. For analyses, the ELISA method using the Titramax-1000 (Heidolph Instruments GmbH & Co., Schwabach, Germany) device was chosen. Blood analyses were conducted at three control points. First: prior to the operation, after the catheterization of the central vein. Second: within 5 min after the completion of CPB. Third: 16 h after the operation ended. Blood samples were collected from the central venous catheter in the internal jugular vein.

### 2.5. Postoperative Delirium

In our research, the clinical indicator of neurovascular unit (NVU) damage was identified as the occurrence and intensity of postoperative delirium, measured via the Cornell Assessment for Pediatric Delirium (CAPD), which has been validated specifically for pediatric patients undergoing cardiac surgery [35]. A score of 9 or higher on this scale signifies the presence of delirium. Evaluations occurred on the first postoperative day in the Department of Anesthesiology and Resuscitation, ensuring that assessments were conducted only after extubation and during periods of spontaneous breathing via the natural airway. To mitigate potential errors in evaluation due to agitation, assessments were conducted 2 h postextubation. Patients were screened using the Richmond Agitation-Sedation Scale (RASS), and if a score of −4 or −5 was noted, further assessment was deferred until the level of consciousness improved to −3 or higher, thereby eliminating the impact of reduced consciousness [36]. Additionally, preliminary evaluations utilizing pain assessment scales were carried out to account for the possible influence of pain on the CAPD results. The Neonatal Infant Pain Scale (NIPS) was utilized for infants under 1 year old [37], while the Face, Legs, Activity, Cry, and Consolability (FLACC) scale was employed for children aged 1 to 3 years [35]. Scores exceeding 3 on these scales indicated the presence of pain. If pain was found during the evaluation, analgesic measures were implemented, followed by re-evaluation. The assessment of postoperative delirium (POD) was conducted only after confirming the absence of pain.

### 2.6. Statistical Data Analysis

Statistical data analysis was performed using BioStat Pro 5.9.8 software. Given that most of the data did not conform to the normal distribution (Shapiro–Wilk test, *p* < 0.05), non-parametric methods of analysis were applied. Data are presented as median (Me) with upper (Q1) and lower (Q3) quartiles. Comparative analysis of quantitative variables was conducted using the Mann–Whitney U test. For related samples, the Wilcoxon signed-rank test was employed. Correlation analysis was performed using Spearman’s rank correlation coefficient (Rho). Differences were considered statistically significant at *p* < 0.05.

## 3. Results

All studied patients were categorized into one of two groups based on the intraoperative use of red blood cell-containing components from donor blood: Group 1—without transfusion (n = 28) and Group 2—with transfusion (n = 50). There were no cases where the initial fluid resuscitation did not include donor blood components but required their use during cardiopulmonary bypass (CPB). However, three patients required intraoperative transfusion of red blood cell suspension following the completion of CPB. These cases were also classified into Group 2.

Table 1 presents the dynamics of the markers used in the study at various control points. For all markers, except for S-100-β protein and TNF-α, the lowest concentrations were observed before the initiation of the surgical intervention. Notably, at the second control point—after the completion of CPB—the concentrations of all markers were statistically significantly higher than the baseline levels: S-100-β (*p* < 0.000001), NSE (*p* < 0.000001), GFAP (*p* < 0.000001), IL-1β (*p* = 0.00026), IL-6 (*p* < 0.000001), IL-10 (*p* < 0.000001), and TNF-α (*p* = 0.00012). Sixteen hours after the surgical intervention, the serum concentration of S-100-β protein was statistically significantly lower (*p* = 0.0053) than the baseline level. The concentration of the marker TNF-α did not differ significantly between the first and third control points (*p* = 0.438), nor did IL-10 (*p* = 0.713). The GFAP marker showed a tendency toward statistical significance in comparison to the baseline level at the third control point (*p* = 0.062). All other markers, 16 h postsurgery, exhibited differences from baseline levels with a sufficient level of statistical significance.

When dividing all studied patients into two groups based on the occurrence of transfusion during the intraoperative period, a general trend towards higher levels of all markers in the group receiving erythrocyte suspension could be observed. Notably, at the initial stage, the values between the groups did not differ for any of the markers (Table 2).

Regarding markers of cerebral injury, it was found that S-100-β protein had a higher concentration at the second evaluation point—after the completion of cardiopulmonary bypass (CPB)—in patients from the second group, which was confirmed by a sufficient level of statistical significance (509.90 (379.30–871.70) vs. 717.10 (517.90–1195.33)) (*p* = 0.024) (see Figure 2) (Table 3). Neuron-specific enolase (NSE) was elevated among patients who received transfusion at the second evaluation point (17.55 (11.19–26.41) vs. 34.05 (17.06–44.90)) (*p* = 0.023), and it also showed significant differences at the third evaluation point (17.63 (7.43–21.66) vs. 22.23 (10.60–41.17)) (*p* = 0.044) (see Figure 3). Similarly, the glial fibrillary acidic protein (GFAP) demonstrated an increase after CPB (*p* = 0.035); however, it is noteworthy that its higher values in group 2 at 16 h postoperation should be interpreted with caution, as they were not formally considered significant (*p* = 0.062) (see Figure 4).

In the analysis of systemic inflammatory markers, the following findings were obtained. The serum concentration of interleukin-1 beta (IL-1β) was significantly higher in the transfusion group (*p* = 0.011) after CPB and exhibited a trend towards a similar difference the following day (16 h postoperation), though without sufficient statistical significance (*p* = 0.060). The systemic inflammatory marker IL-6 showed differences only at the postoperative stage in the third evaluation point (*p* = 0.021). A similar pattern was observed for IL-10, which had higher values in the transfusion group at the third evaluation point (*p* = 0.018). The tumor necrosis factor-alpha (TNF-α) levels were elevated in patients from group 2 only at the measurement taken immediately after the completion of CPB (*p* = 0.011).

When considering the influence of SIRS on neuroinflammation, one would expect to find associations between cerebral injury markers and SIRS markers. The analysis revealed the following statistically significant correlations.

During the intraoperative period, after the CPB, the S-100-β protein correlated with TNF-α with a correlation coefficient of Rho = 0.23 (*p* = 0.04762). NSE exhibited a moderate correlation with TNF-α (Rho = 0.30; *p* = 0.0071) and also showed a correlation with IL-6 (Rho = 0.35; *p* = 0.00206).

At the observation point 16 h after the surgical intervention, the concentration of S-100-β protein displayed a moderate correlation with IL-1β (Rho = 0.32; *p* = 0.00474) and a weak correlation with TNF-α (Rho = 0.28; *p* = 0.00474). On the other hand, NSE demonstrated a strong correlation with TNF-α (Rho = 0.43; *p* = 0.00010) and a notable correlation with blood concentrations of IL-6 (Rho = 0.50; *p* = 0.00001).

To examine the relationship between the use of components of donor blood during the intraoperative period and the concentration of SIRS markers, a correlational analysis was conducted. This analysis revealed the following statistically significant correlations at the second evaluation point: IL-1β—Rho = 0.27 (*p* = 0.01913); TNF-α—Rho = 0.27 (*p* = 0.01599). At the third point of measuring SIRS marker concentrations, a correlation with transfusion was noted for IL-6 with Rho = 0.24 (*p* = 0.03951) and IL-10 with Rho = 0.25 (*p* = 0.02626). According to the Cheddock scale for assessing the strength of association, all identified correlations were considered weak, as their Rho values did not exceed 0.3.

Regarding the markers of cerebral injury, only trends towards statistically significant correlation can be noted (Table 4). For instance, S-100-β protein exhibited a Spearman coefficient of 0.22 with *p* = 0.05651 at the point after the completion of cardiopulmonary bypass (CPB). Similarly, NSE showed Rho = 0.22 with *p* = 0.06386 during the same time frame, and a similar pattern was observed for GFAP with Rho = 0.22 and *p* = 0.05716.

To assess the impact of the total volume of intraoperative transfusions in the second group of patients, it was necessary to analyze how much red blood cell concentrate was administered per kilogram of body weight. The median value was 22.73 (14.40–29.82) mL/kg. In analyzing the correlation between the studied markers and the total volume of transfusion per kilogram of body weight, it was found that IL-1β, according to the Cheddock scale for assessing correlational strength, exhibited a moderate correlation both at the second evaluation point (Rho = 0.37; *p* = 0.010) and at the third point (Rho = 0.33; *p* = 0.023). TNF-α also demonstrated a moderate correlation with the volume of transfusion, but only at the point after the completion of cardiopulmonary bypass (CPB) (Rho = 0.31; *p* = 0.034). Analysis of other SIRS markers, IL-6 and IL-10, did not reveal statistically significant correlations.

Among the markers of cerebral injury, the S-100-β protein showed a moderate correlation with transfusion at both the second (Rho = 0.48; *p* = 0.00065) and third evaluation points (Rho = 0.36; *p* = 0.01330). Neuron-specific enolase (NSE) exhibited a similar pattern: Rho = 0.41 (*p* = 0.00421) after the completion of CPB and Rho = 0.35 (*p* = 0.01667) 16 h after the surgical intervention. The glial fibrillary acidic protein (GFAP) did not show a statistically significant correlation with the volume of transfusion.

Thus, a relationship was established, demonstrating an increase in the concentration of most markers with an increase in the volume of transfusion per kilogram of body weight. All statistically significant correlations are illustrated in Figure 5.

The clinical assessment of neurovascular unit damage in our study was presented via postoperative delirium (POD). When comparing the two groups, it was found that among patients who received transfusions, POD was observed in 19 cases (39.6%), while in the second group, the number of patients diagnosed with delirium was six (20%). The groups did not show a statistically significant difference in the frequency of POD development (*p* = 0.071). However, when analyzing the scores on the CAPD scale in both groups, the following significant difference was revealed: 6 (4–8) points in patients without transfusions compared to 9.5 (7–12) points among patients who underwent intraoperative transfusions (*p* = 0.0056). Furthermore, a correlation was identified between the use of transfusions and the number of points on the CAPD scale (Rho = 0.32; *p* = 0.0042).

No other neurological symptoms, apart from delirium, were recorded in any of the patients.

To exclude the influence of other perioperative factors apart from transfusion, a logistic regression analysis was conducted, which revealed that a statistically significant regression coefficient was observed only for the volume of transfusion per kg of body weight. The coefficient of its impact on the CAPD score was 0.18 (*p* < 0.000001), while the coefficient for its effect on the concentration of the marker NSE was 0.22 (*p* < 0.000001).

## 4. Discussion

### 4.1. Dynamics of Cerebral Injury Markers

From the data presented in Table 1, it can be observed that the highest concentrations for most of the investigated markers were recorded at the second evaluation point, which is logical based on the mechanism of their appearance in the bloodstream. The S-100-β protein is specific to the brain, is located in astrocytes, and enters the bloodstream only when the blood–brain barrier (BBB) is damaged; its serum concentration correlates with the extent of neuronal injury [31,32]. According to the literature, its maximum concentration is reached by the end of cardiopulmonary bypass (CPB) and then decreases sharply in the absence of destructive factors affecting the brain, which corresponds to our findings. Considering that the half-life of S-100-β is approximately 2 h [38], it is understandable that its level in the blood of patients decreases to below baseline levels after 16 h. The second marker of cerebral injury is also strictly tissue-specific for neurons, as it represents an intracellular enzyme, and an increase in its level indicates the presence of cerebral damage [33,39]. Research has shown that the peak concentration of neuron-specific enolase (NSE) is achieved at the end of CPB and 6 h after its completion [38], after which it gradually declines. In our study, the maximum concentration of this marker was recorded immediately after the completion of cardiopulmonary bypass (CPB). A second peak in its concentration could not be observed, as the next serum collection point was 16 h postoperatively, at which time the terminal phase of NSE decline from its maximum level could be assessed. Nevertheless, at this control point, the NSE level remained higher than the preoperative baseline, which aligns with findings from other studies on this marker in cardiac surgery [38,39].

The final marker of brain injury was GFAP, which is specific to astrocytes; an increase in its level in the patient’s blood indicates a disruption in the integrity of the blood–brain barrier (BBB) [40] and is also associated with postoperative cognitive impairments [41]. Our analysis revealed that the peak concentration of GFAP across all control points occurred immediately after the completion of CPB, coinciding with a period of maximum pathological stress on the patient’s brain. However, these findings do not correspond with data from other similar studies, in which the peak concentration of GFAP was noted at 24 [42] and even 72 h [43] after the onset of surgery.

It is likely that future studies in this area would benefit from an extended time interval for the monitoring of these markers, allowing for a more comprehensive understanding of the dynamics of their concentration changes.

### 4.2. Dynamics of SIRS Markers

Regarding the markers of SIRS, our study investigated four markers. The first of these was IL-1, which encompasses a family that includes IL-1α, IL-1β, and IL-1 receptor antagonist (IL-1RN). In our research, we utilized IL-1β, known for its pro-inflammatory effects and short half-life [44]. These characteristics explain the dynamics observed in our data; the peak concentration of the marker was noted after the completion of CPB, coinciding with the period of greatest pathological impact during the entire surgical procedure. Subsequently, a decline in concentration was observed, returning to preoperative levels, which aligns with findings from other similar studies [45].

IL-6 serves as a marker of SIRS with dual pro-inflammatory and anti-inflammatory effects, produced by a variety of cells, including cardiomyocytes, fibroblasts, and endothelial cells [46,47]. Analysis of IL-6 revealed an increase during the CPB period, followed by a further increase on the subsequent day. According to the literature, the peak concentration of this marker is typically reached 24 h postoperatively [48], which is consistent with our findings in the third control point.

IL-10 is an anti-inflammatory cytokine, and it is known that elevated levels of IL-10 in the postoperative period are associated with immunosuppression and an increased risk of infectious complications [49]. According to our data, the peak concentration of this marker was observed at the end of cardiopulmonary bypass (CPB), which is consistent with findings from other similar studies [50]. Following this peak, there was a rapid decline in concentration over 16 h, approaching nearly the initial preoperative levels. Thus, for the studied group of patients, the risk of the described complications and immunosuppression was minimal, which can be attributed to the short duration of CPB, with which IL-10 has been shown to correlate in the findings of Gorjipour et al. [51].

The dynamics of TNF-α observed in our study were similar to those reported in other studies examining this marker in pediatric cardiac surgery [50]. The peak concentration of TNF-α, like that of IL-1β and IL-10, was noted at the second control point, followed by a decrease to baseline levels at the third point. This is logical, considering its relatively short half-life and direct relationship with the damaging factors of the procedure, of which CPB is the primary contributor.

Given the favorable outcomes for all patients in the study and the uncomplicated, brief resuscitation period they experienced, our findings corroborate the results presented by Gorjipour et al. and de Fontnouvelle et al. [51,52]. Their data indicated a correlation between high concentrations of TNF-α and a longer duration of mechanical ventilation in patients undergoing surgical correction of congenital heart defects, a pattern that was not observed in the patients of our study, whose marker levels were at preoperative levels 16 h after the intervention.

### 4.3. The Importance of Transfusion in the Development of Cerebral Damage

When discussing the significance of transfusion in the development of cerebral injury, it is important to highlight the data presented in Table 2. According to these data, the extent of brain injury was significantly higher in the group receiving red blood cell suspension, considering the higher levels of all three specific markers (S-100-β protein, NSE, and GFAP). However, certain limitations of our study should be noted. Firstly, the groups differed in terms of patient body weight as Group 1 had a mean weight of 10.3 kg (8–12), while Group 2 had a mean weight of 8 kg (5.5–9.6) (*p* = 0.0008). The need for transfusion was predominantly determined by body weight, as our clinic has only been able to perform CPB in children without donor blood components starting from a weight of 7.5 kg. Furthermore, although the difference was not significant, Groups 1 and 2 differed in terms of CPB duration. Group 1 had a mean duration of 43 min (38–50), while Group 2 had a mean duration of 50 min (40–63) (*p* = 0.045). No significant difference was observed in the duration of aortic clamping where Group 1 had a mean duration of 27 min (21–33), while Group 2 had a mean duration of 31 min (25.5–42.5) (*p* = 0.064). These aspects could theoretically potentiate a greater release of SIRS markers in Group 2, as noted in our results. Nevertheless, considering the theory that both direct damaging factors and SIRS are sources of neuroinflammation [20,53], there should exist a correlation between SIRS markers and cerebral injury markers. The correlation we observed was present both in the postoperative phase and 16 h after surgery. Thus, it is confirmed that the main factor contributing to the development of neuroinflammation in the studied patients is the SIRS. Evidence that transfusion is a contributing factor lies in the correlation between the occurrence of transfusion and the examined markers. Furthermore, the correlation coefficient was even higher between the dose of intraoperative transfusion per kilogram of patient weight and the markers. The primary cause of the initiation of systemic inflammation during transfusion is the transfer of leukocytes to the patient along with donor blood. However, research indicates that leukoreduction does not completely resolve the issue, as residual donor leukocytes, cytokines, and chemokines continue to stimulate the recipient’s immune response [54]. Furthermore, recent studies have identified several additional mechanisms of immunomodulation associated with transfusion. These include not only the interaction of donor erythrocytes with the recipient’s leukocytes but also the interaction of donor cells with the recipient’s vascular endothelium, which triggers the release of cytokines and activates systemic inflammation [55]. There is evidence indicating a genetic predisposition of donor cells via the CD40L gene to an increased risk of eliciting an immune response during transfusion [56]. Our results align with other studies that have demonstrated the impact of transfusion on the development of cerebral injury and delirium in pediatric cardiac surgery [54,55]. Given all the negative consequences of transfusion, which are also evidenced in our study, investigations aimed at minimizing the use of donor blood components and exploring various aspects of their effects on the patient’s body seem promising [56,57,58,59]. Furthermore, there is a need for a new approach to the diagnosis of postoperative cognitive deficits. It is likely that children who received transfusions during the intraoperative or postoperative period require more frequent testing to identify such deficits and to initiate cognitive rehabilitation in a timely manner.

### 4.4. Advantage of Research

A significant advantage of this study was its novelty, as we were unable to find any literature that investigated the relationship between intraoperative transfusion and brain injury with such strict inclusion criteria. Most studies were focused on cerebral injury during cardiac surgeries in children and the sample consisted of a wide range of patients with all types of congenital heart defects. Another advantage of our study was the use of a wide range of cerebral specific markers, however, this was also a limitation of our work.

### 4.5. Limitations of the Study

The main limitation of our study was the small sample of patients, 78 children, which was due to strict inclusion criteria. Nevertheless, we were able to identify some correlations and confirm the relationship between transfusion and cerebral injury. It is likely that the small sample size contributed to our inability to identify a significant effect of transfusion and its dosage per kg on delirium and serum markers. However, through logistic regression, we were able to control for the influence of other perioperative factors. A second limitation of the study was the small number of control points for investigating blood concentrations of brain injury markers. This is due to the fact that each analysis carries the risk of decreasing hemoglobin levels in the blood, considering the small patient weight. Therefore, in the interest of the children’s safety, we chose these three specific control points. However, for this reason, we were unable for complete investigation of dynamics of the blood markers, considering that their concentration peak may sometimes occur beyond the 16-h period. Ultimately, it is important to consider the potential pathology of the central nervous system that may not have been identified in the patient but could have influenced the neurological outcomes following surgery. Additionally, one must take into account the possible genetic predisposition of certain patients to a heightened inflammatory response.

## 5. Conclusions

When analyzing the dynamics of cerebral damage markers (S-100-β, NSE, GFAP) and SIRS (IL-1b, IL-6, IL-10, TNF-α), we found that the peak concentration occurred at the second control point (the time immediately after the completion of CPB) for all the markers studied, except for IL-6, in which the maximum concentration was observed 16 h after surgery. When comparing the group with the use of transfusion and with refusal from it, statistically higher indicators of cerebral damage markers were revealed at the stage after the completion of CPB for S-100-β, NSE and GFAP and 16 h after surgery for GFAP. The SIRS markers—IL-1 and TNF-α—were significantly higher after the completion of CPB, and IL-6 and IL-10–16 h after surgery. In addition, we noted a correlation between the severity of cerebral damage and SIRS with the fact of the use of donor blood components and their dose per kg of the patient`s body weight. These data confirm that intraoperative transfusion may be a reason for increased levels of SIRS and neuroinflammation with further cerebral damage in children with surgical correction of septal CHD. Such data underscore the importance of developing algorithms for the timely identification and treatment of cognitive impairments following surgery for the correction of CHD in children.

## Figures and Tables

**Figure 1 jcm-13-06050-f001:**
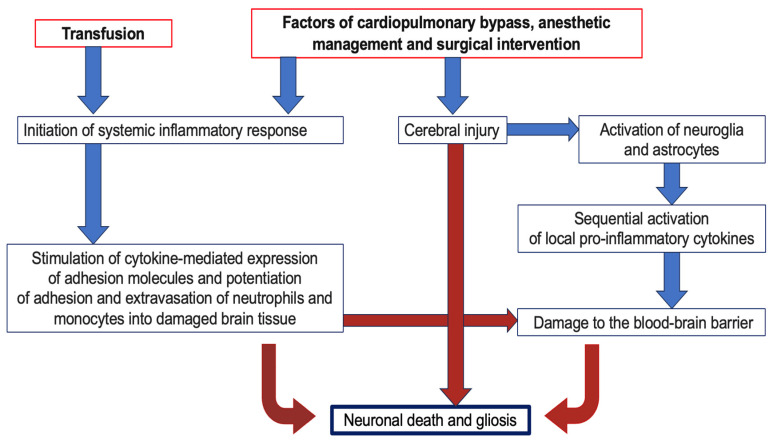
Diagram of the effects of pathological factors on the neurovascular unit during cardiac surgery.

**Figure 2 jcm-13-06050-f002:**
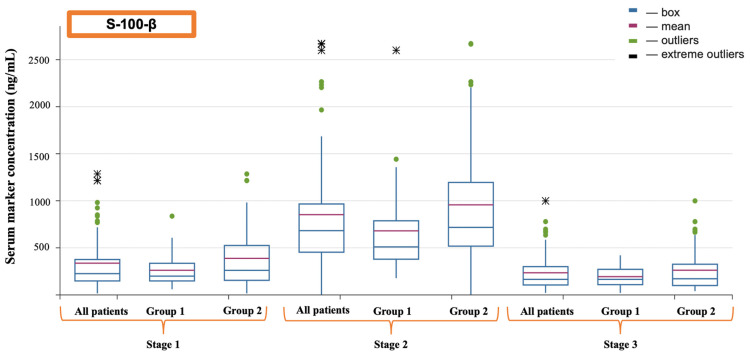
Dynamics of S-100-β.

**Figure 3 jcm-13-06050-f003:**
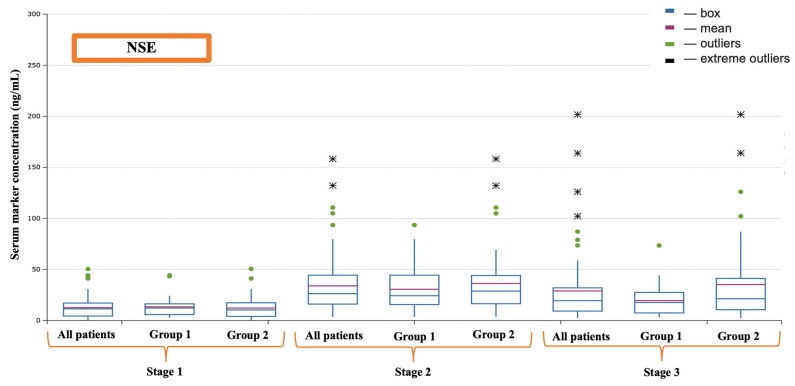
Dynamics of NSE.

**Figure 4 jcm-13-06050-f004:**
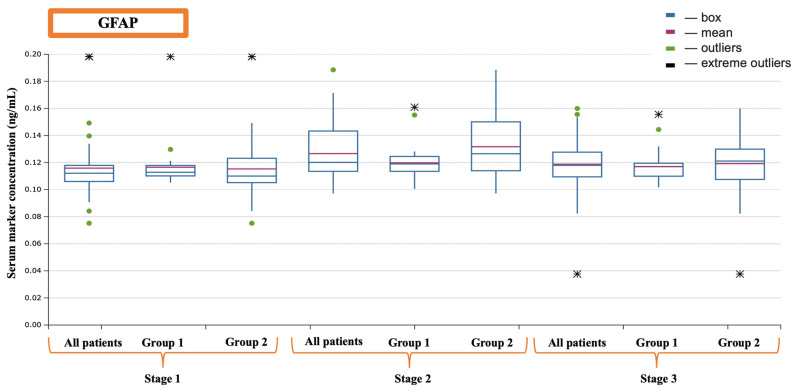
Dynamics of GFAP.

**Figure 5 jcm-13-06050-f005:**
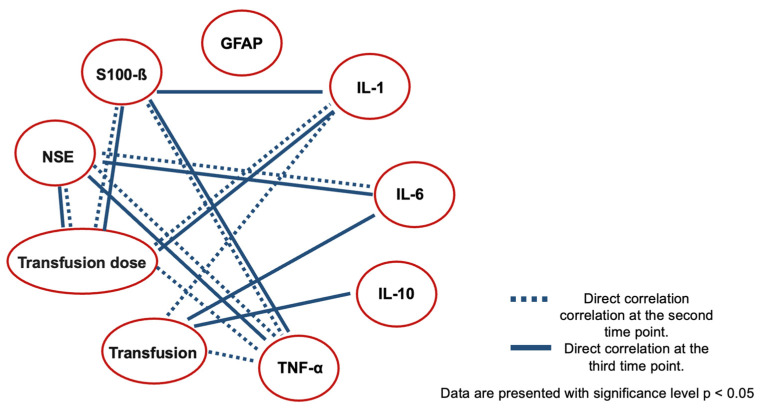
Statistically significant correlations of the markers.

**Table 1 jcm-13-06050-t001:** Dynamics of markers by control points.

Marker	Stages		
	Before the Operation	After the Completion of Cardiopulmonary Bypass	16 h after Surgery
S-100-β (ng/mL)	226.10 (145.00–376.10)	682.00 (453.50–965.65)	171.40 (110.12–314.91)
NSE (ng/mL)	11.42 (4.33–17.10)	23.30 (11.19–39.64)	19.67 (9.15–31.94)
GFAP (ng/mL)	0.1121 (0.1061–0.1212)	0.1215 (0.1135–0.1433)	0.1181 (0.1095–0.1231)
IL-1b (pg/mL)	3.11 (2.55–3.67)	3.50 (3.01–4.40)	3.17 (2.66–4.01)
IL-6 (pg/mL)	2.10 (0.93–2.55)	10.75 (3.38–23.61)	16.18 (3.10–27.18)
IL-10 (pg/mL)	0.79 (0.62–1.28)	7.65 (3.03–13.60)	0.94 (0.70–1.79)
TNF-α (pg/mL)	1.14 (0.86–1.29)	1.33 (1.10–2.10)	1.13 (0.97–1.45)

Note: Table 1 presents the characteristics of the studied Groups 1 and 2, which were formed based on the occurrence of transfusion. Patients in both groups were comparable in terms of the ratio of male to female patients, type of congenital heart defect (CHD), type of surgical access, and duration of aortic clamping. However, the groups differed statistically significantly in age, height, weight, and duration of cardiopulmonary bypass (CPB). Furthermore, patients differed in hemoglobin levels at all stages of the operation; the concentration was higher before the surgery in Group 1, but during CPB and at the end of the operation, the ratio shifted in favor of Group 2. In terms of oxygen transport parameters, such as venous blood saturation, lactate levels, and cerebral oximetry, the groups did not differ statistically significantly, nor did the number of patients requiring inotropic support.

**Table 2 jcm-13-06050-t002:** Characteristics of patients in Group 1 and Group 2.

Investigated Characteristic	Group 1 (n = 30)	Group 2 (n = 48)	*p*
Male, n	11	20	0.6607
Female, n	19	28	
Age (months)	15 (12–28)	11 (8–18)	0.0064
Body weight (kg)	10.3 (8–12)	8 (5.5–9.6)	0.0008
Height (cm)	81 (75.75–90.5)	71 (60.75–78.5)	0.0002
Diagnosis, n (%)			0.1053
ASD, n	20	23	
VSD, n	10	25	
Surgical approach, n (%)			0.3723
Median sternotomy	19	35	
Side sternotomy	11	13	
CPB duration (min.)	43 (38–50)	50 (40–63)	0.0450
Duration of aortic clamping (min.)	27 (21–33)	31 (25.5–42.5)	0.0640
Laboratory indicators			
Hemoglobin before the operation (g/L)	123 (115–128)	113 (108.5–117.5)	0.00003
Hemoglobin level during the CPB (g/L)	87 (81–91.5)	89 (85–98.5)	0.0528
Hemoglobin level at the end of the operation (g/L)	105 (100–110.5)	123 (112–134)	0.00001
Venous blood saturation during the CPB (%)	71 (67–73.25)	69.5 (65–73.5)	0.1990
Venous blood saturation at the end of the operation (%)	71 (68.5–75)	70 (65–77)	0.1891
Blood lactate during the CPB (mmol/L)	1.3 (1.1–1.55)	1.3 (1.15–1.4)	0.3459
Blood lactate at the end of the operation (mmol/L)	1.3 (1.15–1.5)	1.3 (1.0–1.5)	0.2602
Monitoring indicators			
rSO2 indicators before the operation (%)	77 (72–80)	75 (70–78)	0.0568
rSO2 indicators during the CPB (%)	79 (77–81.25)	77 (74–82)	0.0835
rSO2 indicators at the end of the operation (%)	78 (75–82.5)	77 (74–80)	0.0894
Inotropic drugs			
Number of patients with inotropic drugs	11	27	0.0923
Echocardiographic data			
Ejection fraction (%)	65 (63–67)	65 (62–67)	0.49
End-systolic size of left ventricle (mm)	13 (11–16)	14 (10.5–16.5)	0.47
End-diastolic size of left ventricle (mm)	23 (19.5–30.5)	24 (19.5–30.5)	0.69
Left atrial size (mm)	17 (14–20)	18 (14.75–19)	0.43

Note: “during the CPB”—5 min after initiation of CPB; “at the end of the operation”—5 min before the end of surgery.

**Table 3 jcm-13-06050-t003:** Dynamics of markers by control points in groups.

Marker	Stage	Group 1 (n = 30)	Group 2 (n = 48)	*p*
S-100-β (ng/mL)	Before the operation	186.90 (141.70–336.00)	260.80 (154.85–459.71)	0.084
	After the completion of cardiopulmonary bypass	509.90 (379.30–871.70)	717.10 (517.90–1195.33)	0.024
	16 h after surgery	165.40 (141.00–271.90)	183.45 (106.06–347.65)	0.247
NSE (ng/mL)	Before the operation	11.89 (5.90–16.31)	10.36 (4.03–17.34)	0.245
	After the completion of cardiopulmonary bypass	17.55 (11.19–26.41)	34.05 (17.06–44.90)	0.023
	16 h after surgery	17.63 (7.43–21.66)	22.23 (10.60–41.17)	0.044
GFAP (ng/mL)	Before the operation	0.1128 (0.1101–0.1178)	0.1102 (0.1052–0.1232)	0.134
	After the completion of cardiopulmonary bypass	0.1190 (0.1135–0.1245)	0.1231 (0.1138–0.1493)	0.035
	16 h after surgery	0.1156 (0.1099–0.1193)	0.1211 (0.1083–0.1299)	0.062
IL-1b (pg/mL)	Before the operation	2.97 (2.52–3.66)	3.17 (2.56–3.70)	0.436
	After the completion of cardiopulmonary bypass	3.13 (2.88–3.71)	3.88 (3.23–4.63)	0.011
	16 h after surgery	3.04 (2.64–3.49)	3.3 (2.71–4.06)	0.060
IL-6 (pg/mL)	Before the operation	2.14 (1.08–2.47)	2.11 (0.89–2.63)	0.451
	After the completion of cardiopulmonary bypass	12.28 (3.06–17.41)	9.07 (3.43–27.16)	0.231
	16 h after surgery	11.92 (2.21–18.88)	18.97 (4.45–38.25)	0.021
IL-10 (pg/mL)	Before the operation	0.66 (0.59–1.04)	0.88 (0.62–1.29)	0.11
	After the completion of cardiopulmonary bypass	6.32 (2.71–12.20)	8.29 (3.55–15.95)	0.123
	16 h after surgery	0.82 (0.62–1.46)	1.03 (0.45–1.89)	0.018
TNF-α (pg/mL)	Before the operation	1.19 (0.93–1.28)	1.04 (0.83–1.28)	0.126
	After the completion of cardiopulmonary bypass	1.26 (1.04–1.34)	1.52 (1.15–12.22)	0.011
	16 h after surgery	1.11 (0.98–1.24)	1.18 (1.01–1.95)	0.104

**Table 4 jcm-13-06050-t004:** Correlation of markers of cerebral damage and markers of SIRS.

Correlation of Markers at the Second Stage			
	S-100-β Protein	NSE	GFAP
IL-1b	0.14176 (*p* = 0.21569)	−0.14474 (*p* = 0.20611)	0.03819 (*p* = 0.73990)
IL-6	0.09605 (*p* = 0.40284)	0.35 (*p* = 0.00206) *	−0.04037 (*p* = 0.72566)
IL-10	−0.00108 (*p* = 0.99250)	0.05147 (*p* = 0.65450)	0.02675 (*p* = 0.81616)
TNF-α	0.23 (*p* = 0.04762) *	0.30 (*p* = 0.0071) *	0.06155 (*p* = 0.59244)
Correlation of markers at the third stage			
	S-100-β protein	NSE	GFAP
IL-1b	0.32 (*p* = 0.00474) *	−0.15412 (*p* = 0.17789)	0.15316 (*p* = 0.18066)
IL-6	0.03485 (*p* = 0.76193)	0.50 (*p* = 0.00001) *	0.11572 (*p* = 0.31302)
IL-10	0.12187 (*p* = 0.28782)	−0.04421 (*p* = 0.70072)	0.10806 (*p* = 0.34632)
TNF-α	0.28 (*p* = 0.00474) *	0.43 (*p* = 0.00010) *	0.21933 (*p* = 0.05530)

Note: * - statistically significant differences (*p* < 0.05).

## Data Availability

The data presented in this study are available on request from the corresponding author.

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
