# Peer review of "Impact of Intraoperative Blood Transfusion on Cerebral Injury in Pediatric Patients Undergoing Congenital Septal Heart Defect Surgery"

_jcm, 2024, doi:10.3390/jcm13206050_

Round 1

Reviewer 1 Report

Comments and Suggestions for Authors

Upon reviewing the manuscript titled "The relationship between intraoperative transfusion and cerebral injury during surgical correction of congenital heart defects in children", I offer the following suggestions for improvement:

Title

The title accurately reflects the study’s content, but it can be refined for clarity. Consider rewording it to: “Impact of Intraoperative Blood Transfusion on Cerebral Injury in Pediatric Patients Undergoing Congenital Heart Defect Surgery”. This is more concise and directly communicates the study’s focus.

Introduction

The introduction provides a good foundation for the study, but several improvements could enhance its clarity and relevance:

Expand the discussion about cerebral injury in pediatric heart surgery, focusing on why it is a critical concern. Although the introduction touches on postoperative cognitive disturbances, further elaboration on the long-term effects and their clinical significance in young children would strengthen the rationale for the study.

explanation of how systemic inflammatory response syndrome (SIRS) contributes to neuroinflammation and cerebral injury is a key part of the hypothesis. However, this section could be expanded to discuss the specific mechanisms by which transfusion exacerbates SIRS and neuroinflammation in pediatric patients. More detailed references to existing literature could contextualize the study within the broader body of research.

Although the rationale for focusing on transfusion is implied, it could be made more explicit. Explain why there is a need for more specific research on the impact of transfusion on cerebral injury, particularly in pediatric cardiac surgery.

Methods

The methodology section is generally well-structured but could be enhanced with additional detail to improve transparency and reproducibility:

The authors mention that the sample size of 78 is based on calculations from a formula, but the justification could be clearer. Explain why this sample size is sufficient despite the target of 196 patients, and provide a brief statement on how increasing the sample to 200 will improve the study’s power.

The inclusion and exclusion criteria are not clearly detailed. Were children with specific comorbidities, such as neurological conditions, excluded? Clarifying the criteria for participant selection would improve the study’s generalizability and relevance to clinical practice.

The grouping of patients into transfusion and non-transfusion categories is explained, but it might be helpful to elaborate on how patients were assigned to these groups. Were there any predefined criteria for assigning patients to one group or another, or was it based solely on intraoperative necessity?

The biomarkers chosen for measuring cerebral injury and systemic inflammation are appropriate, but the rationale for their selection should be stated explicitly. Why were S-100β, NSE, and GFAP chosen over other potential markers of cerebral damage? Similarly, explain why the study focuses on IL-1β, IL-6, IL-10, and TNF-α for SIRS.

Results

The results section presents a comprehensive analysis of the biomarkers, but it could be improved with the following:

The results are presented in a somewhat dense format. Consider restructuring some sections to make the key findings stand out. For example, using figures or graphs to illustrate the dynamics of biomarkers over time could provide clearer visual representations of the data.

 The statistical methods employed are appropriate, but confidence intervals should be provided alongside p-values for key findings. This would give readers a better understanding of the precision of the estimates.

Discussion

The discussion section offers valuable interpretations but could be expanded and organized more coherently:

While the authors provide a good interpretation of the relationship between transfusion and cerebral injury, this could be further elaborated. Discuss in more detail how these results contribute to existing knowledge and how they could influence clinical decision-making in the context of minimizing transfusions during pediatric cardiac surgery.

While the authors suggest that transfusion exacerbates SIRS, more detail is needed on the mechanisms by which blood components might trigger neuroinflammation. Citing recent studies that discuss the immunomodulatory effects of transfused blood might add depth to the discussion.

The potential clinical applications of this study’s findings are briefly mentioned but should be expanded. How could these results change current transfusion practices? Furthermore, consider discussing how these findings could influence post-surgical monitoring protocols to better identify cerebral injury early.

Although the study’s limitations are acknowledged, this section could be more comprehensive. Discuss the potential confounding factors more explicitly, such as differences in CPB duration or other intraoperative variables. Additionally, consider mentioning the possible impact of unmeasured variables, such as genetic predispositions to inflammation or neurodevelopmental outcomes, which could influence the results.

The authors mention plans to expand the sample size, but the discussion would benefit from more specific suggestions for future research. For example, exploring the effects of alternative transfusion strategies or pharmacologic interventions that could mitigate neuroinflammation would be valuable next steps.

Conclusion

The conclusion effectively summarizes the findings but should emphasize the practical implications of the study more strongly. A clear statement on how these findings can be used to guide clinical practice would strengthen the manuscript’s impact. Additionally, highlighting the importance of future research on non-invasive markers of cerebral injury would add to the conclusion’s depth.

Minor concerns

Several sentences are long and complex, making them difficult to read. Simplifying the sentence structure and ensuring clarity throughout the manuscript would improve readability.

Ensure consistency in the use of terms like “SIRS,” “cerebral injury,” and “transfusion.” Define acronyms such as CPB and POD upon first use.

In summary, the manuscript offers valuable insights into the relationship between transfusions and cerebral injury in pediatric heart surgery patients. With improvements to the clarity of the methods and results, a deeper discussion of the mechanisms, and stronger conclusions regarding clinical relevance, the manuscript would make a more significant contribution to the field.

Comments on the Quality of English Language

 Minor editing of English language required.

Author Response

Comments 1: The title accurately reflects the study’s content, but it can be refined for clarity. Consider rewording it to: “Impact of Intraoperative Blood Transfusion on Cerebral Injury in Pediatric Patients Undergoing Congenital Heart Defect Surgery”. This is more concise and directly communicates the study’s focus.

Response 1: Agree. Thank you for pointing this out. We changed the title of the article. 

Comments 2: Expand the discussion about cerebral injury in pediatric heart surgery, focusing on why it is a critical concern. Although the introduction touches on postoperative cognitive disturbances, further elaboration on the long-term effects and their clinical significance in young children would strengthen the rationale for the study.

Response 2: Agree. We have modified this fragment to emphasize this point.

Comments 3: Explanation of how systemic inflammatory response syndrome (SIRS) contributes to neuroinflammation and cerebral injury is a key part of the hypothesis. However, this section could be expanded to discuss the specific mechanisms by which transfusion exacerbates SIRS and neuroinflammation in pediatric patients. More detailed references to existing literature could contextualize the study within the broader body of research.

Response 3: Thank you for pointing this out. We have added this information.

Comments 4: Although the rationale for focusing on transfusion is implied, it could be made more explicit. Explain why there is a need for more specific research on the impact of transfusion on cerebral injury, particularly in pediatric cardiac surgery.

Response 4: Agree. We have modified this fragment to emphasize this point.

Comments 5: The authors mention that the sample size of 78 is based on calculations from a formula, but the justification could be clearer. Explain why this sample size is sufficient despite the target of 196 patients, and provide a brief statement on how increasing the sample to 200 will improve the study’s power.

Response 5: Agree. We have modified this fragment to emphasize this point.

Comments 6: The inclusion and exclusion criteria are not clearly detailed. Were children with specific comorbidities, such as neurological conditions, excluded? Clarifying the criteria for participant selection would improve the study’s generalizability and relevance to clinical practice.

Response 6: We agree with this comment. We have added the inclusion and exclusion criteria.

Comments 7: The grouping of patients into transfusion and non-transfusion categories is explained, but it might be helpful to elaborate on how patients were assigned to these groups. Were there any predefined criteria for assigning patients to one group or another, or was it based solely on intraoperative necessity?

Response 7: Thank you for pointing this out. We have added this information.

Comments 8: The biomarkers chosen for measuring cerebral injury and systemic inflammation are appropriate, but the rationale for their selection should be stated explicitly. Why were S-100β, NSE, and GFAP chosen over other potential markers of cerebral damage? Similarly, explain why the study focuses on IL-1β, IL-6, IL-10, and TNF-α for SIRS.

Response 8: Agree. We have added this information.

Comments 9: The results are presented in a somewhat dense format. Consider restructuring some sections to make the key findings stand out. For example, using figures or graphs to illustrate the dynamics of biomarkers over time could provide clearer visual representations of the data.

Response 9: We presented the dynamics of cerebral markers in graphs. We believe that this is sufficient to avoid overloading the article with illustrations.

Comments 10: The statistical methods employed are appropriate, but confidence intervals should be provided alongside p-values for key findings. This would give readers a better understanding of the precision of the estimates.

Response 10: Agree. We have modified this fragment to emphasize this point.

Comments 11: While the authors provide a good interpretation of the relationship between transfusion and cerebral injury, this could be further elaborated. Discuss in more detail how these results contribute to existing knowledge and how they could influence clinical decision-making in the context of minimizing transfusions during pediatric cardiac surgery.

Response 11: Agree. We have modified this fragment to emphasize this point.

Comments 12: While the authors suggest that transfusion exacerbates SIRS, more detail is needed on the mechanisms by which blood components might trigger neuroinflammation. Citing recent studies that discuss the immunomodulatory effects of transfused blood might add depth to the discussion.

Response 12: Agree. We have added the information about immunomodulatory effects of transfused blood.

Comments 13: The potential clinical applications of this study’s findings are briefly mentioned but should be expanded. How could these results change current transfusion practices? Furthermore, consider discussing how these findings could influence post-surgical monitoring protocols to better identify cerebral injury early.

Response 13: Agree. We have added this information.

Comments 14: Although the study’s limitations are acknowledged, this section could be more comprehensive. Discuss the potential confounding factors more explicitly, such as differences in CPB duration or other intraoperative variables. Additionally, consider mentioning the possible impact of unmeasured variables, such as genetic predispositions to inflammation or neurodevelopmental outcomes, which could influence the results.

Response 14: Thank you for pointing this out. We have modified this fragment to emphasize this point.

Comments 15: The authors mention plans to expand the sample size, but the discussion would benefit from more specific suggestions for future research. For example, exploring the effects of alternative transfusion strategies or pharmacologic interventions that could mitigate neuroinflammation would be valuable next steps.

Response 15: Agree. We have modified this fragment to emphasize this point.

Comments 16: The conclusion effectively summarizes the findings but should emphasize the practical implications of the study more strongly. A clear statement on how these findings can be used to guide clinical practice would strengthen the manuscript’s impact. Additionally, highlighting the importance of future research on non-invasive markers of cerebral injury would add to the conclusion’s depth.

Response 16: Agree. We have modified this fragment to emphasize this point.

Comments 17: Several sentences are long and complex, making them difficult to read. Simplifying the sentence structure and ensuring clarity throughout the manuscript would improve readability.

Ensure consistency in the use of terms like “SIRS,” “cerebral injury,” and “transfusion.” Define acronyms such as CPB and POD upon first use.

Response 17: We have fixed these errors.

Reviewer 2 Report

Comments and Suggestions for Authors

This is a nice study with clear and robust message. Basically the authors do a great work explaining that blood transfusion may cause cerebral injury in children undergoing a surgical correction of heart defects. My main criticism is that based on table 2 there are extreme differences between the 2 groups. 

Group that received transfusion was : younger, Lower weight, higher CBD duration, higher aortic clamping etch. The claim that you are making is that the changes in the biomarkers of brain injury is due to the blood transfusion. Someone though can argue that the change is because in general those children were sicker, and therefore needed the transfusion for there unstable condition which therefore is the real cause of the cerebral injury. 

This study would be great if the authors performed propensity score matching and showing the same results. If they cannot proceed with propensity score matching, then a multivariate regression adjusting for all this confounders and proving that the increase in the biomarkers is due to the blood transfusion would be acceptable. If none of the above are performed, it would be hard to pin this association to the blood transfusion, and therefore the paper becomes solely descriptive- losing a lot of interest and impact. 

Comments on the Quality of English Language

In general this is a well written work. My main concerns revolve around analysis rather than presentation and writing. 

Author Response

Comments 1: Group that received transfusion was : younger, Lower weight, higher CBD duration, higher aortic clamping etch. The claim that you are making is that the changes in the biomarkers of brain injury is due to the blood transfusion. Someone though can argue that the change is because in general those children were sicker, and therefore needed the transfusion for there unstable condition which therefore is the real cause of the cerebral injury. 

Response 1: Agree. Thank you for pointing this out. We conducted a logistic regression analysis that revealed transfusion to be a significant factor associated with the development of cerebral injury. This information has been included in the Results section.

Reviewer 3 Report

Comments and Suggestions for Authors

Thank you to the authors for submitting their manuscript to our journal. The article focuses on the relationship between intraoperative transfusion and cerebral injury during the surgical correction of congenital heart defects in children. The following minor revisions are required to enhance clarity and specificity:

1. In the abstract, it is necessary to include the Background subsection by referencing it in line 8 before "the components." This addition will provide context and facilitate understanding.

2. The introduction must define the term “neuronal brain injury” as used within the text, ensuring that readers comprehend its specific implications.

3. We advise the creation of a graphical abstract that succinctly summarizes the findings and key messages of the study, facilitating quick comprehension of the results.

4. The manuscript should clarify whether the enrolled patients underwent brain imaging techniques, such as MRI, before or after the intervention and if they exhibited any postoperative neurological symptoms.

5. It is essential to specify in the title that the study pertains to patients with atrial septal defects or ventricular septal defects, improving the accuracy of the study's scope.

6. Additionally, the authors should describe how the shunt magnitude was evaluated (e.g., transthoracic echocardiogram, transesophageal echocardiogram, transcranial Doppler).

7. Finally, please include echocardiographic data in the descriptive table of the sample, including ejection fraction, end-systolic volume, end-diastolic volume, and left atrial volume.

These revisions will enhance the manuscript’s clarity and improve its contribution to the field.

Author Response

Comments 1: In the abstract, it is necessary to include the Background subsection by referencing it in line 8 before "the components." This addition will provide context and facilitate understanding.\

Response 1: Agree. We have modified this fragment to emphasize this point.

Comments 2: The introduction must define the term “neuronal brain injury” as used within the text, ensuring that readers comprehend its specific implications.

Response 2: We have fixed these errors.

Comments 3: We advise the creation of a graphical abstract that succinctly summarizes the findings and key messages of the study, facilitating quick comprehension of the results.

Response 3: Agree, but we presented the dynamics of cerebral markers in graphs. We believe that this is sufficient to avoid overloading the article with illustrations.

Comments 4: The manuscript should clarify whether the enrolled patients underwent brain imaging techniques, such as MRI, before or after the intervention and if they exhibited any postoperative neurological symptoms.

Response 4: Agree. We have added this information in materials and methods.

Comments 5: It is essential to specify in the title that the study pertains to patients with atrial septal defects or ventricular septal defects, improving the accuracy of the study's scope.

Response 5: Agree. Thank you for pointing this out. We changed the title of the article. 

Comments 6: Additionally, the authors should describe how the shunt magnitude was evaluated (e.g., transthoracic echocardiogram, transesophageal echocardiogram, transcranial Doppler).

Response 6: Agree. We have added this information.

Comments 7: Finally, please include echocardiographic data in the descriptive table of the sample, including ejection fraction, end-systolic volume, end-diastolic volume, and left atrial volume.

Response 7: Agree. We have added this information.

Round 2

Reviewer 1 Report

Comments and Suggestions for Authors

The manuscript has been adequately revised